# Enhanced Surface Plasmon by Clusters in TiO_2_-Ag Composite

**DOI:** 10.3390/ma15217519

**Published:** 2022-10-26

**Authors:** Yongjun Zhang, Zhen Xu, Shengjun Wu, Aonan Zhu, Xiaoyu Zhao, Yaxin Wang

**Affiliations:** 1School of Material and Environmental Engineering, Hangzhou Dianzi University, Hangzhou 310018, China; 2Department of Clinical Laboratories, Sir Run Run Shaw Hospital, School of Medicine, Zhejiang University, Hangzhou 310020, China; 3College of Chemistry, Nankai University, Tianjin 300071, China

**Keywords:** TiO_2_-Ag composite, nanocaps decorated by clusters, SERS

## Abstract

The surface plasmon in the composite composed of the noble metals and the semiconductors is interesting because of the various charges and the potential applications in many fields. Based on a highly ordered 2D polystyrene spheres array, the ordered composite nanocap arrays composed of TiO_2_ and Ag were prepared by the co-sputtering technique, and the surface morphology was tuned by changing TiO_2_ sputtering power. When TiO_2_ sputtering power was 60 W and Ag sputtering power was 10 W, the composite unit arrays showed the nanocap shapes decorated by many composite clusters around. The composite clusters led to the additional local coupling of the electromagnetic fields and significant Surface-Enhanced Raman Scattering (SERS) observations, which was also confirmed by the finite-different time-domain simulation. The SERS-active substrate composed of the composite nanocaps decorated by clusters realized the accurate detection of the thiram with concentrations down to 10^−9^ M.

## 1. Instruction

Pesticides are often used to decrease plants diseases, insect pests, and improve plant growth in modern agriculture. When the pesticides are abused with the exceeding levels, the chemicals remain in the environment and accumulate in food products, which finally are harmful to human health [1]. As a widely used antibacterial pesticide, thiram is shown to accumulate in human body and harm human skin and mucous membranes. Although the detailed regulations are approved regarding the thiram use, the rapid and accurate detections of thiram residues challenge the ordinary consumers [2,3].

Surface plasmon-related observations in nanostructures have recently attracted many researchers due to both the theoretical importance and the wide applications [4]. In the view of the classic physics [5,6], the coherent movements of the electrons driven by the electromagnetic field of incident light carry significant energy, which can be delivered to their surroundings by the relaxation process, resulting in the local energy enhancement around the nanostructures [7,8]. When the nanostructures are near to each other, the local electromagnetic fields tend to couple together, which leads to the significant enhancement of the localized energy on the nanometer scale named as hot spots [9,10,11]. The energy and the distribution of hot spots depend on many parameters of the nanostructures, such as the materials, the shapes, the chemical compositions, and the separations between the nanostructures, which are also the tunable choices to get the desired hot spots [12,13]. To utilize the conventional information technology, the ordered units composed of semiconductor materials arouse extensive interests, which can be integrated onto the chips. In addition to lithography and the electron beam lithography (EBL), the nanotechnology based on the nanosphere lithography is more favorable due to its easy and cheap operations [14,15]. Furthermore, the nanosphere lithography can prepare the nanostructures with the unusual properties beyond lithography and the EBL, such as metal film over nanosphere (MFON) surfaces. In addition to the noble metals, some semiconductors are also a good choice for SERS-active substrate, because the semiconductors contribute to the enhanced electromagnetic field owing to the charge transfer process [16,17]. However, the enhancement factor from the semiconductors is usually small because the charge density is low compared to the noble metals [18,19,20]. Therefore, the researchers prefer the composite of the semiconductors and the noble metals, in which the noble metals work as the electron reservoir and the semiconductors work as the donors of the different chargers [21]. Therefore, the exciting observations are reported for the composites composed of the noble metals and the semiconductors for the synergetic properties [22,23]. In the study of the composite of the semiconductors and the noble metals, ZnO and TiO_2_ are the preferred choices due to the large energy gap and cheap availability. Jiwei reported a remarkable enhancement of Raman scattering achieved by submicron sized spherical ZnO superstructures, in which Mie resonances and the highly efficient charge transfer (CT) contribution enhance the SERS signals together [24,25]. Li’s group confirms the spillover of hydrogen species and its role in tuning the activity and selectivity in catalytic hydrogenation in Au/TiO_2_/Pt sandwich nanostructures by in situ SERS study reveals that hydrogen species can efficiently spillover at Pt-TiO_2_-Au interfaces, which provides molecular insights to deepen the understanding of hydrogen activation and boosts the design of active and selective catalysts for hydrogenation [26]. Our investigations show the surface plasma resonance peaks have a blue shift when the TiO_2_ thickness changed for bilayer film TiO_2_/Ag on two-dimensional (2D) polystyrene (PS) bead array, which is because the intermediate electrical properties of metal and semiconductor composite systems have changed. Compared with Ag and TiO_2_ bilayer films, Ag and TiO_2_ co-sputtering can reduce the aggregation of silver nanoparticles [27,28]. In addition, the enhanced roughness is observed for the same time. The surface morphology and the roughness have significant effects on SERS behaviors as known to all. Based on the observations mentioned above, the effects of TiO_2_ contents on the surface morphologies and the SERS behaviors are investigated by co-sputtering TiO_2_ and Ag onto 2D PS beads array. Scanning Electron Microscope (SEM) and Transmission Electron Microscope (TEM) images show the TiO_2_-Ag composite arrays of the rough surfaces form on 2D PS beads array, and the surface roughness changes with TiO_2_ content by changing TiO_2_ sputtering power. When 60 W sputtering power is applied, the large surface roughness is obtained with some cluster around the nanocap, which leads to the significant SERS effects because of the additional coupling formation. 

## 2. Experiment

The 4-mercaptobenzoic acid (4-MBA) was purchased from Sigma-Aldrich Co., Ltd. (Shanghai, China). The solution of polystyrene bead with a diameter of 500 nm was obtained from Duke Co., Ltd. (Palo Alto, CA, USA). with a concentration of 10 wt%. Ag and TiO_2_ targets were bought from Beijing TIANRY Science and Technology Developing Center (99.99 wt%, Beijing, China). Silicon wafers were obtained from Hefei Kejing Materials Technology Co., Ltd. (Hefei, China). The deionized water (18.2 MΩ·cm^−1^) was prepared from a Millipore water purification system (Shanghai, China). PS monolayer film was prepared by means of self-assembly as reported on silicon wafer (Kejing, Hefei, China) according to our previous research. The silicon wafer was immersed in the mixed solution of ammonia (Sigma-Aldrich, Shanghai, China), hydrogen peroxide (Sigma-Aldrich, Shanghai, China), and deionized water to get the hydrophilic surface, which was boiled it for 20 min. One hundred micrometers polystyrene bead solution and 70 μL absolute ethanol (Sigma-Aldrich, Shanghai, China) were sonicated (XinZhi, Ningbo, China) for 10 min to get completely mixed solution. Then, the mixed solution was applied to the deionized water surface to form a dense monolayer. Clean silicon wafers were used to pick up dense monolayers. The model of the magnetron control sputtering system is Silver (TianQi, Beijing, China) and titanium dioxide (TianQi, Beijing, China) sputtered films were deposited in ATC 1800-F (Shenyang, China). The base pressure is 3∙10^−6^ Pa and the pressure of experiment is 0.6 Pa during sputtering. The sputtering power of the target TiO_2_ is 20–60 W and the sputtering power of 10 W is applied to the target Ag. The morphology and microstructure were measured on scanning electron microscope (SEM) with model JEOL 6500F (JEOL, Tokyo, Japan) and transmission electron microscope (TEM) with model JEM-2100H (JEOL, Tokyo, Japan). X-ray diffraction (XRD) was performed on the Rigaku D/MAX 3C X-ray diffractometer with Cu Kα radiation (λ = 1.5418 Å) (Waltham, MA, USA). UV-3600 spectrophotometer (Shimadzu, Kyoto, Japan) was taken to obtain the UV-Vis spectrum. The Raman spectrum was collected on Renishaw Raman (Renishaw, London, UK) with a spectral resolution of 1 cm^−1^.

## 3. Results and Discussion

As shown in Figure 1, TiO_2_ and Ag are sputtered from TiO_2_ target and Ag target, respectively. This experimental set-up can tune the deposition rates by changing the sputtering powers of TiO_2_ and Ag. The properties of the composites are affected by the content of TiO_2_, which are affected by sputtering power in the deposition process.

When the sputtering power is 20 W for TiO_2_, the film is mainly composed of Ag due to the slow deposition rate of TiO_2_ and the film morphology is akin to the Ag film on PS beads as reported earlier (Figure 2a). When the sputtering power is increased to 40 W for TiO_2_, more TiO_2_ is deposited with Ag and the coarse surfaces are observed due to the increased concentration of TiO_2_. Between the TiO_2_ caps, some synaptic structures are observed as shown in Appendix A, which make the gap size narrow. The caps are composed of many nanoparticles of the size around 10 nm (Figure 2b). When the sputtering power is increased to 60 W for TiO_2_, some clusters with the size around 100 nm are observed between the nanocaps, which indicates the increased TiO_2_ content results in the increased sizes of the TiO_2_-Ag nanoparticles (Figure 2c). When TiO_2_ is incorporated into Ag film by co-sputtering, the film shows the rough surface because of the immiscible separation between TiO_2_ and Ag. The increased TiO_2_ content results in the increased roughness due to the enhanced TiO_2_ separation. The large roughness leads to significant particles with decreased sizes, which results in the clusters with the size around 100 nm, leading to additional nanogaps between the nanocaps and the clusters (Appendix A). Figure 2d shows the extinction spectrum of TiO_2_-Ag bilayer film, and two obvious extinction bands can be seen. The two peaks near 400–700 nm and 700–1000 nm can be interpreted as of TiO_2_-Ag composites and the local surface plasmon resonance (LSPR) of simple Ag, respectively. The red shift of extinction peaks and the increase of resonance intensity are observed when TiO_2_ deposition power increases from 20 W to 60 W, which can be ascribed to the decreased Ag sizes and the increased surrounding dielectrics.

TEM images in Appendix A confirm the low concentration of TiO_2_ additions have little effect on Ag nanocaps, and Ag nanoparticles are still compact under TEM observations. When the concentration of TiO_2_ addition is increased by the sputtering power 40 W, some small Ag nanoparticles are created and gather around the nanocaps, which increase the roughness around the nanocaps. When the sputtering power is increased to 60 W, the small TiO_2_-Ag nanoparticles tend to stick to each other, which leads to the large aggregations between the TiO_2_-Ag nanocaps as shown by the red circle. TEM image shows the particles are made of Ag nanoparticles separated by TiO_2_. Ag nanoparticles show the size around 8–10 nm. TiO_2_ layers are around 3nm thickness and TiO_2_ form the matrix, covering the Ag nanoparticles.

Figure 3a shows TEM images and EDS mapping images for the composite TiO_2_-Ag. TEM image shows that the nanocap is decorated by the cluster. High Resolution Transmission Electron Microscope (HRTEM) analysis shows that the size of silver nanoparticles is about 15 nm, TiO_2_ has both crystalline and amorphous states. The results show that compared with pure silver, co-sputtered of Ag and TiO_2_ can effectively inhibit the aggregation of Ag nanoparticles. EDS mapping is carried out for the cluster as guided by the blue square. The red, green, and yellow mapping images correspond to elements silver, oxygen, and titanium, respectively, which show the uniform distributions of Ag, O, and Ti elements in the sample. XRD patterns in Figure 3b show the existences of TiO_2_ and Ag in the composite. The diffraction peaks at 33.6° agree with TiO_2_ (211) in JCPDS card 33-1381, indicating TiO_2_ in rutile phase. Ag diffraction peaks at 38.1° and 44.3° can be assigned to the face-centered cubic crystal structure (111) and (200), in agreement with JCPDS card 04-0783.

SERS properties of different substrates were tested by using 4-MBA with the concentration of 10^−3^ M as the detection molecule. The spectra are collected on a Renishaw Raman system with HeCd laser irradiation (633 nm), laser power 17 mW, the laser power attenuation 1%. The SERS signals confirm the excellent SERS enhancement performance. SERS signals show the decrease first from deposition power 20 W to 40 W for TiO_2_ deposition, and the SERS signal increases for the deposition power 60 W in Figure 4. 

Employing finite-difference time-domain (FDTD) simulations calculate the hot spot distribution of Ag and TiO_2_ composites. Figure 5a–c shows the ideal model for the sample and the position of the monitor. The black wireframe represents the smallest simulation area, and the transparent plane represents the monitor on the x-y plane. These calculation results (Figure 5d–f) show that the hot spots in the sample are mainly distributed around the Ag-TiO_2_ nanocaps. When TiO_2_ is added into the nanocaps, the couplings between the nanocaps are weaken due to the enhanced dielectrics. When more TiO_2_ is added, many large TiO_2_-Ag clusters result in more hot spots located between the cluster-cluster and cluster-nanocap in addition to the nanocap-nanocap coupling. Therefore, the cluster-surrounded nanocap array show excellent surface plasmon coupling suitable as SERS active substrates.

At present, the social food safety problem is serious. In order to solve the current problem, we used nanocap array modified by Ag and TiO_2_ clusters to detect trace thiram. The latest US Environmental Protection Agency revision require the thiram concentrations below 15 ppm (about 10^−6^ M) [29]. Therefore, the thiram concentration of 10^−6^ M, 10^−7^ M, 10^−8^ M, and 10^−9^ M are applied to our SERS-active substrates. 

Figure 6 shows the characteristic peaks from thiram, the peak at 1380 cm^−1^ from CN stretching vibration mode and CH_3_ symmetry breaking, the peak at 1138 cm^−1^ from CN stretching vibration, and the peak at 562 cm^−1^ from the stretching vibration mode of S–S [30,31,32]. Through deep analysis of the SERS spectrum, it can be found that thiram with a concentration of 10^−9^ M can be detected on this substrate. We selected the intensity of the peak at 1380 cm^−1^ of Raman spectrum for quantitative analysis of thiram, which satisfied the linear relationship of y = 65,500 − 6500x. The correlation coefficient is 0.99, indicating the suitable application for thiram detection.

## 4. Conclusions

In conclusion, the clusters in TiO_2_-Ag composite were prepared by co-sputtering TiO_2_ and Ag onto 2D PS bead array. SEM, TEM, UV-Vis, XRD, and SERS measurements showed the surface morphologies and the surface plasmon coupling depended on the sputtering power of TiO_2_ and Ag. The increased TiO_2_ concentrations resulted in abundant hot spots due to the additional coupling between the clusters and the nanocaps, the clusters and the clusters, when more TiO_2_ was sputtered for 60 W. The SERS-active substrates composed of the TiO_2_-Ag composite nanocap decorated by clusters were utilized for thiram detection, and the accurate detection was realized for the thiram concentration down to 10^−9^ M, which showed the great potential applications in food safety.

## Figures and Tables

**Figure 1 materials-15-07519-f001:**
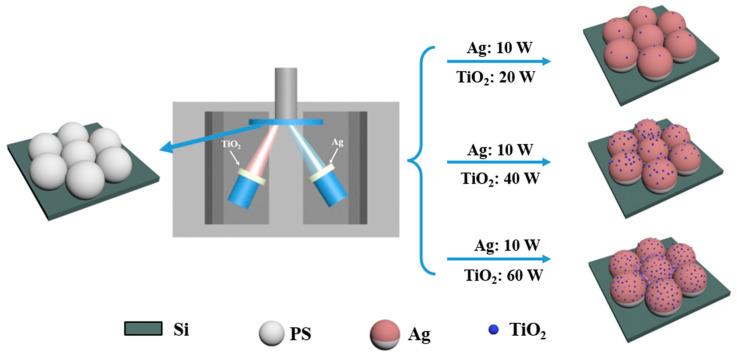
Schematic diagram of deposition of TiO_2_-Ag composites.

**Figure 2 materials-15-07519-f002:**
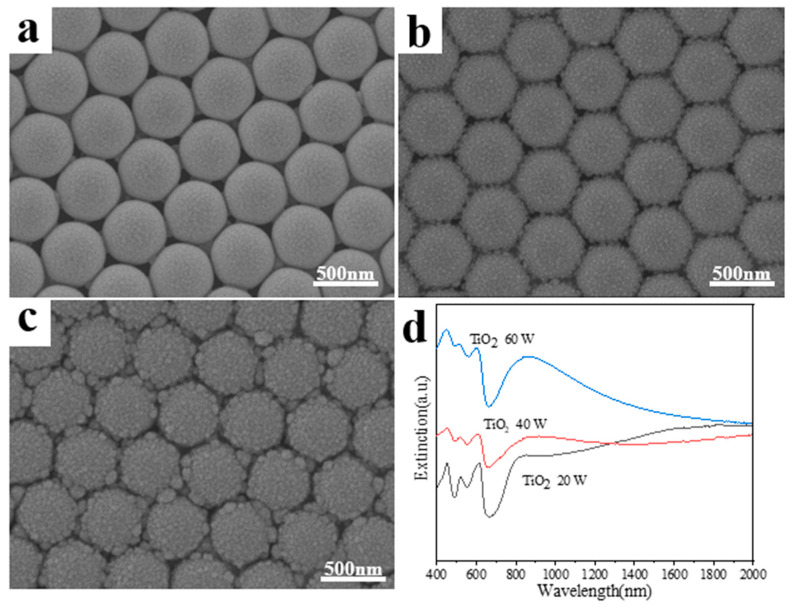
(**a**–**c**) SEM images for the samples with TiO_2_ sputtering power 20 W, 40W and 60 W respectively and Ag sputtering power of remains unchanged; (**d**) Extinction spectra for the above three structures.

**Figure 3 materials-15-07519-f003:**
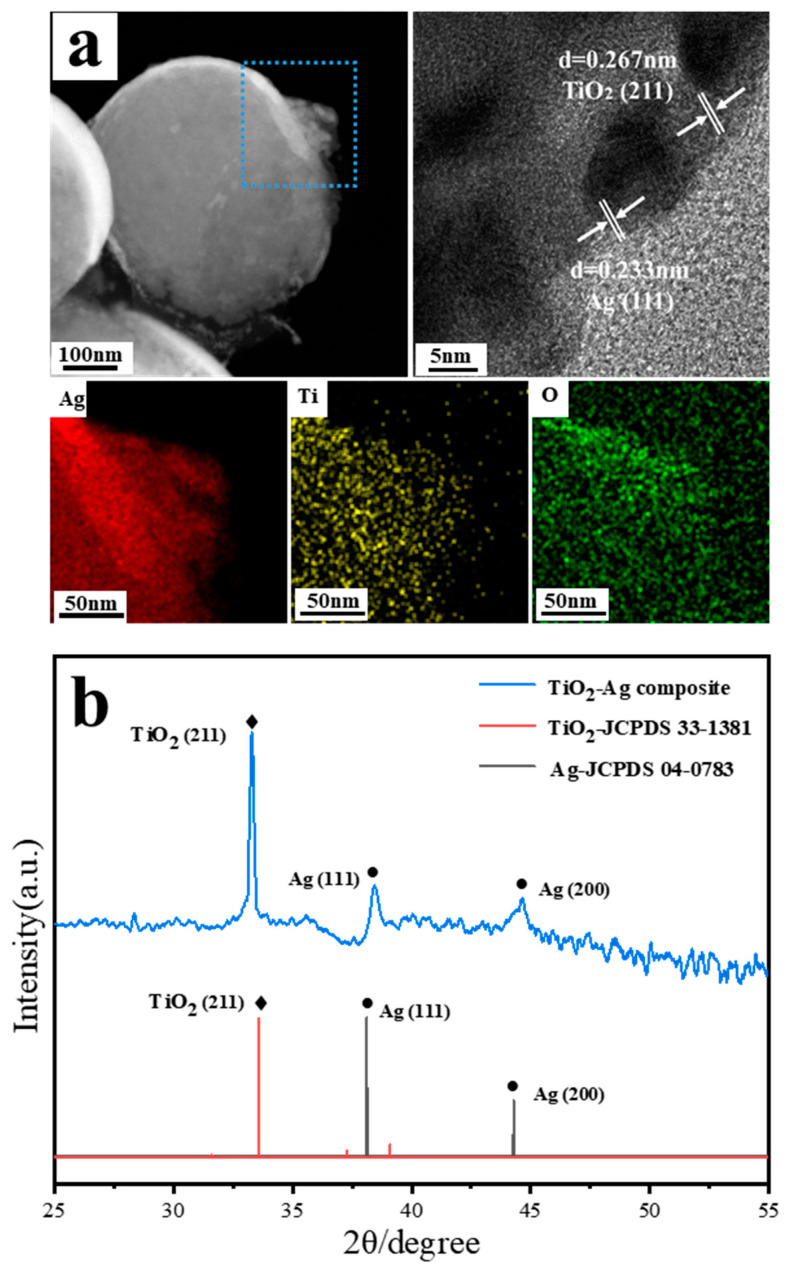
(**a**) TEM image and EDS mapping for the composite TiO_2_-Ag; (**b**) The X-ray diffraction pattern of composites TiO_2_-Ag with TiO_2_ sputtering power 60 W.

**Figure 4 materials-15-07519-f004:**
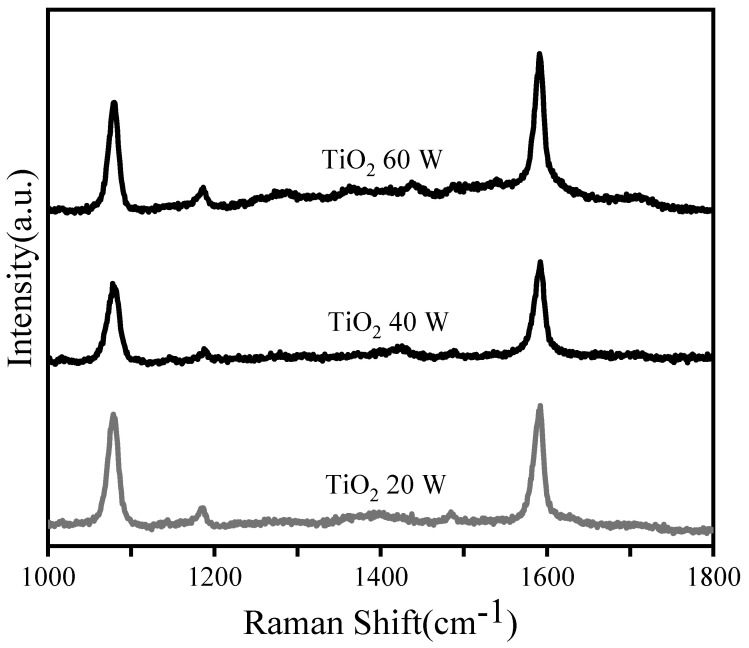
SERS spectra for the 4-MBA probe molecules absorbed on the composite TiO_2_-Ag with Ag sputtering power 10 W and TiO_2_ sputtering powers 20 W, 40 W and 60 W.

**Figure 5 materials-15-07519-f005:**
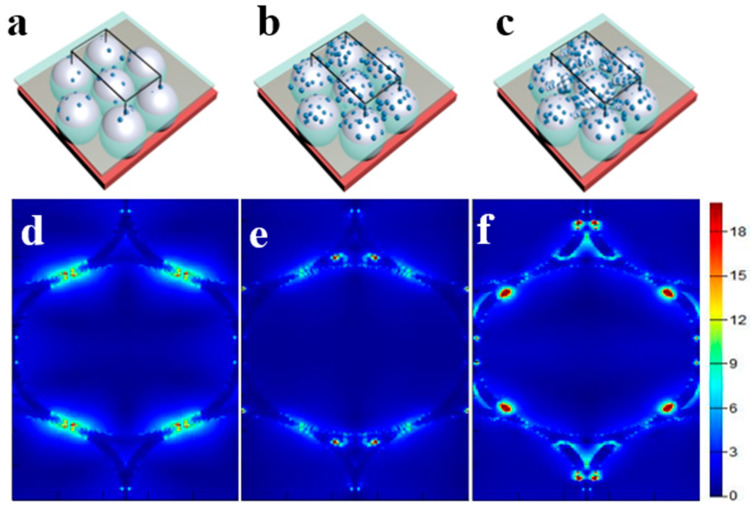
(**a**–**c**) Ideal form of nanocap structure with different TiO_2_ content. Transparent plane represents the monitor on the x-y plane; (**d**–**f**) Electromagnetic field enhancement distribution of three structures.

**Figure 6 materials-15-07519-f006:**
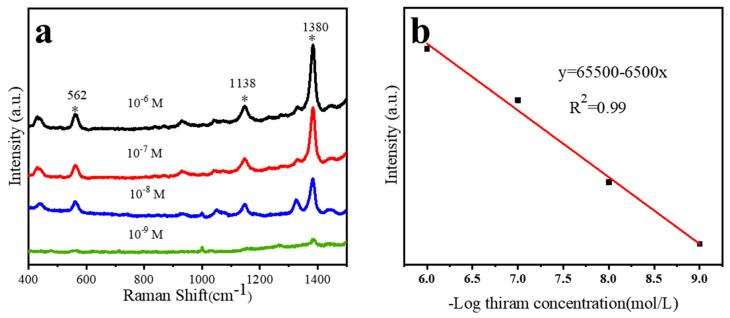
(**a**) SERS spectra of cluster in TiO_2_-Ag composite with different concentrations of thiram (10^−6^–10^−9^ M); The star * highlights the position of the characteristic peak of thiram. (**b**) Take the negative logarithm of the concentration values of thiram with different concentrations as the X value and take the Raman peak at 1380 cm^−1^ as the Y value as a linear function image.

## Data Availability

All the data is available within the manuscript.

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
