# Peer review of "Enhanced Surface Plasmon by Clusters in TiO2-Ag Composite"

_materials, 2022, doi:10.3390/ma15217519_

Round 1
Reviewer 1 Report
Line 91: What adhesion exists between polystyrene and silicon substrate? Why the choice of 500nm in the section of the PS particles?
Line 117: During the deposition, a silver film or silver clusters are made, I would not be sure of the formation of silver nanoparticles, it is not clearly described how the 500nm PS particles are covered. The authors should better explain the morphology obtained
Line 147: From the reading, it is unclear what measure they performed and on what substrate. If the substrate is silicon, is it a measure in the reflection? The location of the localized plasmon of silver is not seen, also given the morphology it is undoubtedly a matter of scattering phenomena. How long do you see in the model, are the polystyrene particles completely covered with silver? To clarify the type of measurement carried out and can a scale on the y axis. From the optical measures carried out, I sincerely see localized plasmonic effects
In fig. 3, Is the sphere's image due to the PS particles? Moreover, I would improve the quality of the X-ray measurements.
Line 201: What it means to apply to our SERS-active substrates? Were they exposed to vapours? How were polluted champions made? Which active sample was used in sensing measurements?
To my concern, the article is to be reviewed with greater reviews before publication.
Reviewer 2 Report
The paper reports on new SERS substrate made depositing TiO2/Ag composite and its use for pesticide detection using SERS. The paper can be published however it has to be revised before doing so following the comments below:
1. The structure is not clear, is it Ag and TiO2 deposited simultaneously at PS spheres?, or is it TiO2 on Ag spheres?, It is not described very well and figure 1 needs more explanation as one sees green spheres of PS and grey spheres of Ag, so which is which????
2. What is the role of TiO2?, why Ag is not enough is unclear, please explain?
3. The LSPR wavelength is around 650nm which agrees with the laser wavelength used for Raman, is this accidental or based on design?, please explain it clearly.
4. Several review references on plasmonics, SERS, ec., are outdated and the authors should be more updated in the newly emerging works, see for example:
Yuqing Yang , Niamh Creedon, Alan O’Riordan and Pierre Lovera, Surface Enhanced Raman Spectroscopy: Applications in Agriculture and Food Safety, Photonics 2021, 8, 568. https://doi.org/10.3390/photonics8120568
Vasyl Shvalya, Gregor Filipič, Janez Zavašnik, Ibrahim Abdulhalim, Uroš Cvelbar, Surface-enhanced Raman spectroscopy for chemical and biological sensing using nanoplasmonics: The relevance of interparticle spacing, and surface morphology, Appl. Phys. Rev. 7, 031307 (2020); https://doi.org/10.1063/5.0015246
Simone Balbinot, Anand Mohan Srivastav, Jasmina Vidic, Ibrahim Abdulhalim, Marisa Manzano, Plasmonic Sensors for Food Control, Trends in Food Science & Technology, 11, 128-140 (2021). https://doi.org/10.1016/j.tifs.2021.02.057.
5. SERS has been used used for other pesticides such as paraoxon, see for example:
A. Baruch-Leshem, S. Isaacs, S. K. Srivastava, I. Abdulhalim, A. Kushmaro, H. Rapaport, Quantitative Assessment of Paraoxon Adsorption to amphiphilic ß-sheet Peptides Presenting the Catalytic Triad of Esterases. J. Colloids. Int. Sci., 530, 328–337 (2018).
Reviewer 3 Report
The manuscript entitled “Enhanced surface plasmon by clusters in TiO2-Ag composite ” has been submitted by the authors. Some issues to be addressed will improve the quality of the manuscript. Therefore, I recommend this work could be published after the major revision
1. The author should write down the novelty of this paper.
2. The English composition requires many improvements. The authors should proofread the manuscript carefully to minimize grammatical errors.
3. All the references mentioned in the paper should be cited in the text or vice-versa.
4. This research topic has been widely studied, and many studies have been performed. The author, please add a comparative table for the reader's clear understanding.
5. Enough references do not support the characterization and result and discussion parts. It may be supported by the recent relevant references (before 2015).
Sci Rep 6, 20103 (2016): Nanoscale, 2013,5, 4427-4435

Round 2
Reviewer 1 Report
To my concern, the paper is publishable in the present form
Author Response
Thank you again for the comments and valuable suggestions on our manuscript entitled " Enhanced surface plasmon by clusters in TiO2-Ag composite" (No.materials-1940994).
Reviewer 2 Report
The authors responded to most of the comments however important comment number 3 is not answered properly. If it is deone by design, then explain how the design was done? how you managed to get the LSPR at the SERS laser wavelength??
Author Response
Thank you again for the comments and valuable suggestions on our manuscript entitled " Enhanced surface plasmon by clusters in TiO2-Ag composite" (No.materials-1940994).
Please see the attachment.

Reviewer 3 Report
The author solved all comments carefully, I recommended accepting in the present form.
Author Response

(The authors gave the same response as above.)
